# Lipid Serum Profiling of Boar-Tainted and Untainted Pigs Using GC×GC–TOFMS: An Exploratory Study

**DOI:** 10.3390/metabo12111111

**Published:** 2022-11-15

**Authors:** Kinjal Bhatt, Thibaut Dejong, Lena M. Dubois, Alice Markey, Nicolas Gengler, José Wavreille, Pierre-Hugues Stefanuto, Jean-François Focant

**Affiliations:** 1Organic and Biological Analytical Chemistry Group (OBiAChem), MolSys, University of Liège, 4000 Liège, Belgium; 2TERRA Teaching and Research Center, Gembloux Agro-Bio Tech, University of Liège, 5030 Gembloux, Belgium; 3Animal Production Unit, Walloon Agricultural Research Centre, 5030 Gembloux, Belgium

**Keywords:** lipidomics, fatty acids, boar taint, gas chromatography, GC×GC–TOFMS

## Abstract

Mass spectrometry (MS)-based techniques, including liquid chromatography coupling, shotgun lipidomics, MS imaging, and ion mobility, are widely used to analyze lipids. However, with enhanced separation capacity and an optimized chemical derivatization approach, comprehensive two-dimensional gas chromatography (GC×GC) can be a powerful tool to investigate some groups of small lipids in the framework of lipidomics. This study describes the optimization of a dedicated two-stage derivatization and extraction process to analyze different saturated and unsaturated fatty acids in plasma by two-dimensional gas chromatography–time-of-flight mass spectrometry (GC×GC–TOFMS) using a full factorial design. The optimized condition has a composite desirability of 0.9159. This optimized sample preparation and chromatographic condition were implemented to differentiate between positive (BT) and negative (UT) boar-tainted pigs based on fatty acid profiling in pig serum using GC×GC–TOFMS. A chemometric screening, including unsupervised (PCA, HCA) and supervised analysis (PLS–DA), as well as univariate analysis (volcano plot), was performed. The results suggested that the concentration of PUFA ω-6 and cholesterol derivatives were significantly increased in BT pigs, whereas SFA and PUFA ω-3 concentrations were increased in UT pigs. The metabolic pathway and quantitative enrichment analysis suggest the significant involvement of linolenic acid metabolism.

## 1. Introduction

Lipidomics, or the comprehensive analysis of lipids, is rapidly expanding and providing critical information to the field of bioscience. Lipids have been studied using mass spectrometry (MS) for decades, but lipidomics is one of the newest members of the “omics” family introduced by Spener [1] and Han and Gross [2]. Although metabolomics mainly focuses on the hydrophilic classes, lipidomics has emerged as an independent omics owing to its structural complexities and hydrophobic and amphiphilic nature, which provides a wide range of biological functions [3,4]. When considering lipidomics, liquid chromatography (LC)–mass spectrometry (MS)-based techniques are widely used. However, enhanced separation capacities and lower limits of detection are challenging for these LC–MS(/MS)-based approaches [4]. Aside from this, chemical derivatization has the potential to make some families of lipids more “gas chromatography (GC)-amenable”, allowing more sensitive GC–MS to also be considered.

The gold standard for lipid extraction techniques for biological matrices based on chloroform and methanol was introduced by Folch [5] and Bligh and Dyer [6]. When considering GC, chemical derivatization is essential for the conversion of the extracted fatty acid components of lipids into more volatile and stable derivatives such as methyl esters to analyze saturated and unsaturated fatty acids. There are six frequently used protocols for the derivatization of lipids in plasma using GC [7]. These are potassium hydroxide (KOH) derivatization, trimethylsulfonium hydroxide (TMSH) derivatization, TMSH direct injections, boron trifluoride (BF_3_) derivatization [8,9,10,11,12], chlorhydric acid (HCl) derivatization, and sodium hydroxide (NaOH) + BF_3_ derivatization. The fatty acid composition of plasma can be divided into four classes: saturated fatty acids (SFA), monounsaturated fatty acids (MUFA), polyunsaturated fatty acids (PUFA (ω-3)), (PUFA (ω-6)), and derivatives. As mentioned, there are two main families of PUFAs: (PUFA (ω-3)) and (PUFA (ω-6)) because of the relevance of PUFAs to human health: polyunsaturated fatty acids (PUFAs) may modulate inflammatory processes and regulate the antioxidant signaling pathway. They impact liver lipid metabolism and physiological responses of other organs, including the heart [13]. Each derivatization technique has pros and cons; however, when selecting a sample preparation protocol, its efficiency over all four classes of fatty acids becomes essential to consider. A systematic comparative study of different derivatizations and extraction efficacies was conducted to determine lipid composition as fatty acid methyl esters (FAMEs) in plasma by Ostermann (Figure 1) [7]. Both the HCl and NaOH+BF_3_ derivatization protocols proved to be suited for the analysis of overall fatty acid patterns without discriminating individual classes of FAMEs. However, as demonstrated by Micalizzi et al. [14], NaOH + BF_3_ derivatization can be fully automated using a dual head autosampler, providing an upper edge compared to HCl derivatization.

Untargeted lipid profiling can help to better understand the ongoing biological mechanisms that have an observable effect, such as the production of detectable smell in food products. Boar taint is a pungent, unpleasant smell or taste found in the meat of some uncastrated male pigs. This smell is caused by a complex mixture of molecules released upon heating the meat [15]. The widely known molecules responsible for boar taint are androstenone and skatole [16,17,18]. The surgical castration of male piglets is a traditional practice to prevent boar taint in meat worldwide. However, it is performed without anesthesia or analgesia, causing great pain to the piglets. Hence, due to increased animal welfare concerns, European pork production stakeholders agreed to prohibit surgical castration of male piglets from 2018. These objectives are yet to be fully achieved successfully [15,19].

This study uses pig serum as a biological sample instead of preferred backfat to analyze boar taint. Currently, boar taint detection techniques used in slaughterhouses are sensory evaluation by the human nose upon heating the fat [20] and spectrophotometric detection at 580 nm. Modern analytical techniques investigated for boar taint identification are UHPLC–HRMS [17], GC–MS [16,21,22], HPLC–FD [23], and Raman spectroscopy [24]. These modern analytical techniques focus on quantification and/or validation of known boar taint compounds, e.g., indole, skatole, and androsterone in porcine adipose tissue [16,17,18,19,20,21,22,23,24].

In this study, we report the two-stage sample preparation protocol for extracting lipids from 25 µL of plasma/serum for GC×GC–TOFMS. The optimized approach becomes valuable for analytes with low abundance, e.g., PUFAs (ω-3). The protocol has been optimized using human plasma, confirmed with NIST plasma metabolites, and implemented on animal serum, indicating its efficiency and usability. Widely available analytical approaches for identifying boar taint in pork mainly focus on androstenone and skatole molecules using the backfat of the pig. With this protocol, we investigated a new approach, focusing on lipids by studying the fatty acid composition of pig serum responsible for boar taint (boar-tainted) (BT) compared to that without boar taint (untainted) (UT). Ultimately, the lipid profiling of pig serum enables the use of a different type of biological sample instead of backfat; thus, different molecules of fatty acids will provide us with new biological information.

## 2. Materials and Methods

### 2.1. Samples and Chemicals

For derivatization, 0.5 M sodium methoxide (CH_3_ONa) and boron trifluoride (BF_3_) 20% solution in methanolic solution were purchased from ACROS organics and Sigma-Aldrich, respectively. A Supelco 37 FAMEs standard mixture was purchased from Sigma-Aldrich. A 10 ppm FAMEs solution was prepared in dichloromethane. The n-alkanes mixture (C7-30) in hexane was purchased from Millipore Sigma and diluted to 10 ppm in hexane for the calculation of linear retention indices (LRIs).

Pooled human plasma of six humans was purchased from TCS Biosciences (Buckingham, UK). Biological reference standard SRM 1950 “Metabolites in frozen human plasma” was purchased from NIST. To optimize extraction and derivatization conditions, pooled human plasma was used and stored as sub-aliquots at −80 °C to avoid thawing effects. For the identification of analytes, 37 FAMEs standard mixture, NIST SRM 1950, and n-alkane standard were analyzed.

The pig blood samples (n = 40) (sex = male, age = 6 months ± 15 days) were collected in 16 × 125 mm BD Vacutainer^®^ SST™ plastic tubes (cat# 367985) to obtain serum. Serum samples identified as boar-tainted (n = 20) and untainted (n = 20) by the human nose at slaughterhouse were stored at −20 °C. The blood samples were collected after the slaughtering of pigs as a part of a large study.

### 2.2. Instrumental Method

GC×GC–TOFMS analysis was performed with a Pegasus 4D (LECO Corporation, St. Joseph, MI, USA) equipped with Agilent 7890 GC. The analysis was performed using a normal column set configuration, Rxi-5Sil-MS (30 m × 0.25 mm ID × 1.0 µm df), and VF-17ms (2 m × 0.25 mm ID × 0.5 µm df). A guard column of 2 m was installed.

The temperature program for the primary and secondary oven was the same, starting at 50 °C and holding for 2 min, then increasing temperature to 160 °C at 30 °C/min, followed by a ramp of 2 °C/min until it reached 280 °C. At last, the 300 °C temperature was achieved with a ramp of 30 °C/min and held for 2 min. The total run time for the GC method was 69.33 min. The secondary oven temperature offset was +5 °C, and the modulator temperature offset was +15 °C. A mass range of 45 to 700 *m*/*z* was collected at an acquisition rate of 150 spectra/s by positive mode electron ionization (EI) at 70 eV. Ion source and transfer line temperatures were maintained at 230 °C and 250 °C, respectively.

### 2.3. Sample Preparation

As shown in Figure 2, 500 µL of CH_3_ONa was added to 25 µL of pooled human plasma for the first stage of derivatization and heated for a specified time (Table 1). After cooling, 500 µL of methanolic BF_3_ solution was added and again heated (Table 1). In the end, 300 µL of heptane was added for liquid–liquid extraction. The upper heptanoic solution was collected and injected into the GC×GC–TOFMS system. Pig serum samples of 25 µL were analyzed using optimized derivatization and extraction protocol.

### 2.4. Data Processing

The data processing for the optimization of the sample preparation conditions was performed in ChromaTOF^®^ (ver. 4.72, LECO Corp., St. Joseph, MI, USA). The putative identification of analytes was conducted with a spectral similarity library search against the NIST17 mass spectral library. Analytes were quantified at specific *m*/*z*: 74, 55, and 67 for FAMEs with zero to two double bonds, respectively, while FAMEs with three to six double bonds were quantified at 79 *m*/*z*. The composite desirability and response optimization plot for the Design of Experiments (DoE) were analyzed on Minitab (Ver. 20.2.0). The pig plasma data were processed using GC Image^TM^ (ver. 2021r). Data pre-processing of normalization to sample median, square root transformation, and mean centering were conducted prior to applying chemometric tools. The chemometric tests, unsupervised screening (PCA, HCA), univariate analysis profile (volcano plot), multivariate supervised analysis (PLS–DA), pathway analysis, and enrichment analysis were performed using MetaboAnalyst 5.0 (Xia Lab, McGill University, Montréal, QC, Canada) [25]. The pathway topology analysis is measured with relative betweenness centrality for the node importance and globaltest for enrichment analysis.

## 3. Results

### 3.1. Optimization of Derivatization and Extraction Conditions via Experimental Design

To improve measurement efficiency and obtain clean chromatographic separation of the lipids, a two-step sample extraction and derivatization approach was optimized using the DoE. An amount of 25 µL of pooled human plasma was used for the optimization process (Figure 2). In the first step, the addition of CH_3_OH helps in the derivatization of fatty acids bound in sources such as cholesterol in plasma (base-catalyzed transesterification). The second step, the addition of BF_3_, helps in the esterification of free fatty acids (acid-catalyzed esterification). Liquid–liquid extraction was performed using heptane at the end of the second derivatization. However, the optimization of temperature (T_1_, T_2_) and time (t_1_, t_2_) for sample preparation can help us achieve overall maximum extraction efficiency. A two-level full factorial design for 16 different conditions with an additional three center points for a total of 19 runs was evaluated (Table 1).

The formation of a structured chromatographic separation gives an advantage over the conventional GC approach. As shown in Figure 3b, the separation of FAMEs in the first dimension occurs as the number of carbon atoms or volatility decreases, while in the second dimension, as the polarity of FAMEs increases, the number of double bonds increases. The elution pattern of FAMEs also depends on the position of double bonds. The parallel-aligned ω compounds are separating at an obtuse angle, with higher-ω FAMEs eluting before lower ones. The increased film thickness of the ^1^D and ^2^D columns (1.0 µm, 0.5 µm) enables structured chromatographic elution without a wrap-around effect by retaining the compound on the ^2^D column longer. Thus, the structured chromatographic separation of FAMEs can become a valuable tool for identifying unknown analytes.

As per the certificate of analysis (CoA) of NIST SRM 1950 (Appendix A), due to the vastly varying concentration range of fatty acids in blood plasma, a number of analytes—C16:0, C18:1 n-9, C18:2 n-6—become oversaturated in the chromatogram. However, it remains possible to achieve the separation of all FAMEs. Therefore, to avoid bias while optimizing the DoE parameters, a representative of each class was selected instead of a summation of the entire individual classes.

For the response optimization, five analytes were selected, covering the different classes of FAMEs, including SFA, MUFA, omega-3, and omega-6 (Figure 3a) (Table 2). The center points injected three times at a randomized interval had an overall %RSD of 8.47, indicating the reproducibility of the method. The optimal composite desirability is 0.9159 for the optimized sample preparation condition (Appendix A). Thus, the optimized derivatization method is efficient in extracting all the classes of FAMEs without creating a bias for a specific class.

### 3.2. Identification of Pigs Responsible for Boar Taint by Lipid Profiling of Serum Using Optimized Derivatization and Separation Conditions

In total, 40 pig serum samples were analyzed using optimized sample preparation and chromatographic conditions, out of which 20 were identified as boar-tainted and 20 as untainted pigs at the slaughterhouse by the human nose. As previously observed in human plasma (Figure 3), it was possible to observe SFA, MUFA, PUFA, and cholesterol derivatives (Figure 4) in pig serum. For a better illustration, the contour plots were reconstructed in Python.

In pig serum samples, out of 39 features, 13 SFAs, 8 MUFAs, 7 PUFAs ω-3, 9 PUFAs ω-6, ω-9, and 2 cholesterol derivatives are present (Appendix A). Unsupervised principal component analysis (PCA) was performed to visualize a potential clustering trend between the boar-tainted and untainted pig serum samples. As seen from the PCA scores plot (Figure 5a), PC 1 and PC 2 contributed 62.4% variance. There is one outlier outside the 95% confidence interval identified by a Grubbs test, possibly because of the less regulated sample collection conditions at a slaughterhouse. A significant clustering trend was observed between the two groups, which indicated that the fatty acid derivatives were able to differentiate the boar-tainted pigs from untainted pigs.

The hierarchical clustering result is shown as a heat map (Figure 6). Using Euclidean distance measure and clustering algorithm Ward.D for the top 25 features, it is shown that SFA and PUFA (ω-3) are present in higher concentrations in untainted pig serum. In contrast, PUFA (ω-6), (ω-9), and cholesterol derivatives are in higher abundance in boar-tainted pig serum.

The volcano plot combines a fold change (FC) analysis and a *t*-test. For this test, the *t*-test threshold was set at 0.05, and the direction of comparison was boar-tainted pig serum divided by untainted pig serum (BT/UT). The volcano plot (Figure 7a) has 11 important features, of which two SFAs are downregulated in boar-tainted pigs, while six PUFAs ω-6 and two cholesterol derivatives are upregulated in boar-tainted pigs. Moreover, the partial least squares discriminant analysis (PLS–DA) had 8 features out of 39 features with a threshold of variable importance in projection (VIP) score > 0.9.

## 4. Discussion

A comprehensive analytical method workflow for analyzing lipids in plasma/serum has been detailed herein. The method includes liquid–liquid extraction of lipids from 25 µL plasma/serum optimized to maintain a wide selectivity towards multiple classes of FAMEs (SFA, MUFA, and PUFA (ω-3 and ω-6)). A micro-volume extraction optimized using pooled human plasma (Figure 3), tested on NIST plasma metabolites (Appendix A), and utilized on pig serum (Figure 4) illustrates that the optimized sample preparation protocol is efficient for both human plasma and animal serum for lipid extraction, opening the possibility to translate this analytical protocol to other plasma/serum-related studies.

GC×GC–TOFMS is a powerful technique for lipidomics as it provides structured chromatographic separation, which adds value to identifying untargeted lipidomics. The optimized method presented here provides valuable insight on FAMEs identification without requiring in-depth MS/MS investigations.

For the first time, lipid profiling of serum for boar-tainted and untainted pigs is analyzed, identifying distinguished FAMEs class composition. The results pertaining to the observed significant presence of PUFA ω-6 and cholesterol derivatives in boar-tainted pig serum are supported by multivariate analysis, while SFA and PUFA ω-3 are significant in untainted pig serum. These differences in the lipid composition are opening new investigation routes to better understand bore taint deviation. Moreover, 36 features were subjected to pathway and enrichment analysis (Figure 8). The linoleic acid metabolism pathway was the key metabolic pathway, with a pathway impact of 1 supported by quantitative enrichment analysis (Figure 8).

A ω-3-rich feed for pigs is recommended to maintain pork as a good nutritional source of ω-3 fatty acids for humans [26]. Moreover, the outcome of lipid profiling of pig serum gives rise to an intriguing possibility of the importance of PUFA (ω-3) not only as a nutritional essential, but also in the involvement of reducing boar taint in pigs.

## Figures and Tables

**Figure 1 metabolites-12-01111-f001:**
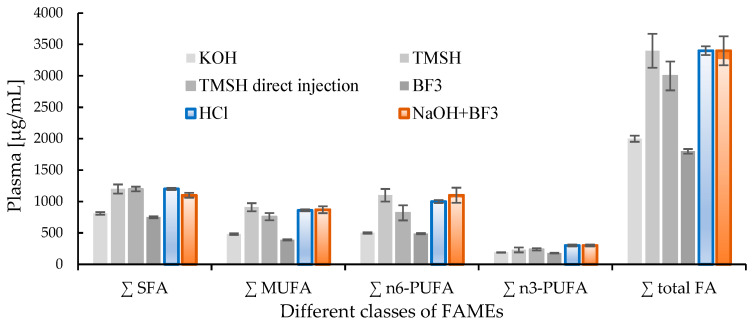
Comparison of derivatization efficacy of different derivatization methods [7].

**Figure 2 metabolites-12-01111-f002:**
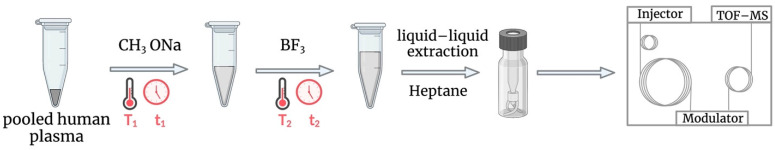
Schematic diagram of two-stage chemical derivatization and extraction approach.

**Figure 3 metabolites-12-01111-f003:**
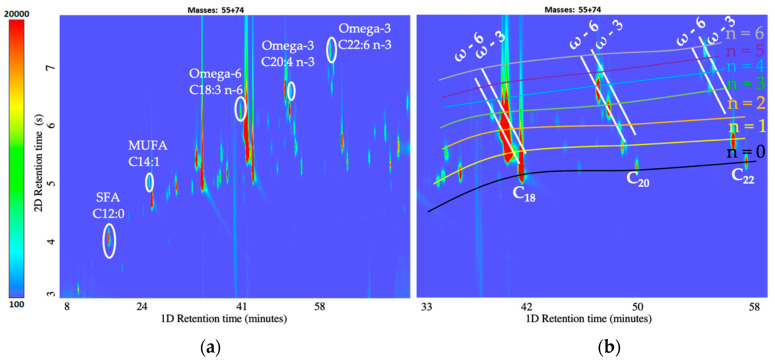
(**a**) Representative of each class selected for response optimization: zoomed-in Contour plot of pooled Human plasma at *m*/*z*: 55 + 74. (**b**) Zoomed-in contour plot of pooled human plasma for C_18_ to C_22_ region at *m*/*z*: 55 + 74.

**Figure 4 metabolites-12-01111-f004:**
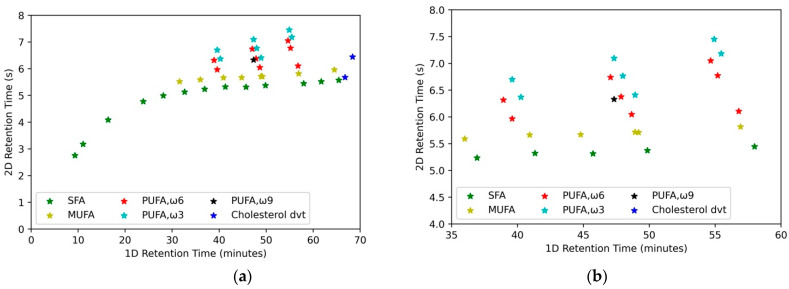
(**a**) Reconstructed contour plot of pig serum. (**b**) Zoomed-in reconstructed contour plot for C_18_ to C_22_ region for pig serum.

**Figure 5 metabolites-12-01111-f005:**
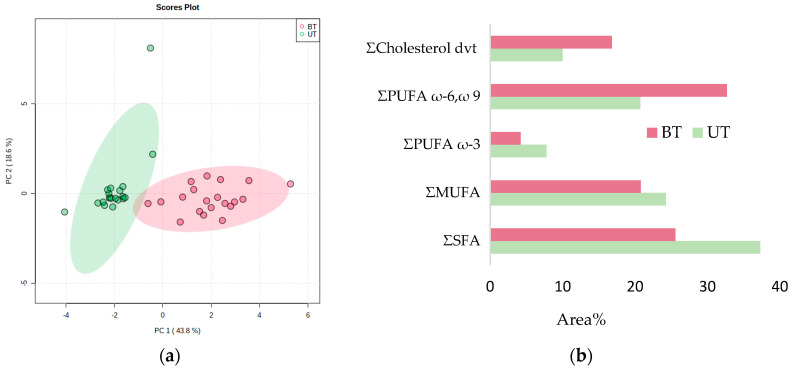
(**a**) PCA score plot. (**b**) % Area contribution of fatty acids per class for boar–tainted (BT) and untainted (UT) pig serum samples.

**Figure 6 metabolites-12-01111-f006:**
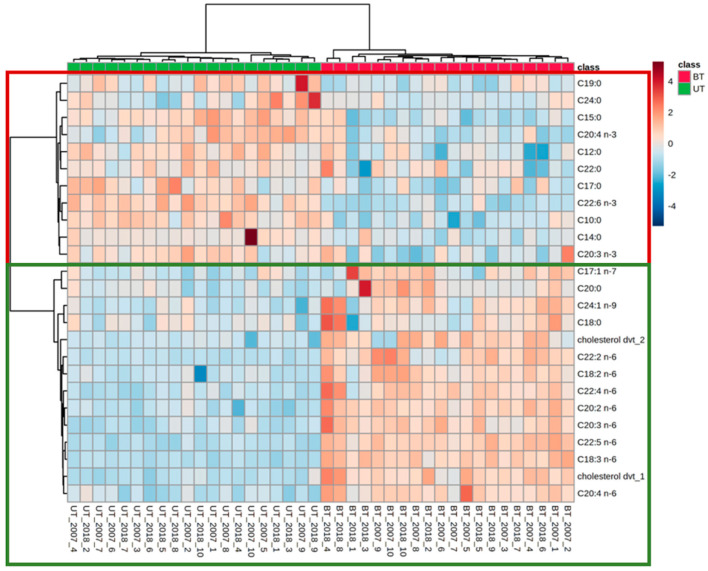
Heat map using top 25 features of boar–tainted (BT) and untainted (UT) pig serum.

**Figure 7 metabolites-12-01111-f007:**
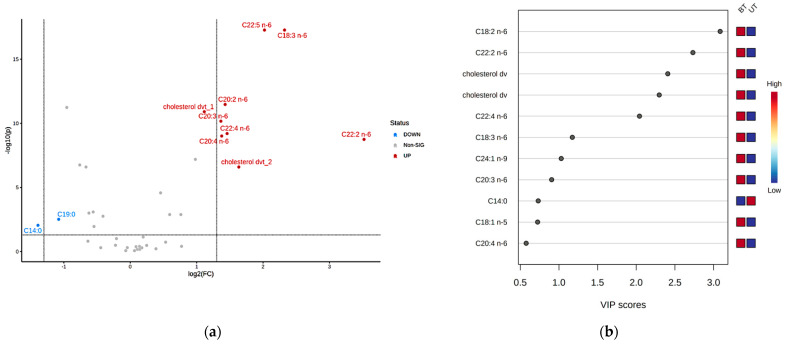
(**a**) Volcano plot of differentially expressed features (BT/UT). (**b**) PLS–DA: VIP score graph.

**Figure 8 metabolites-12-01111-f008:**
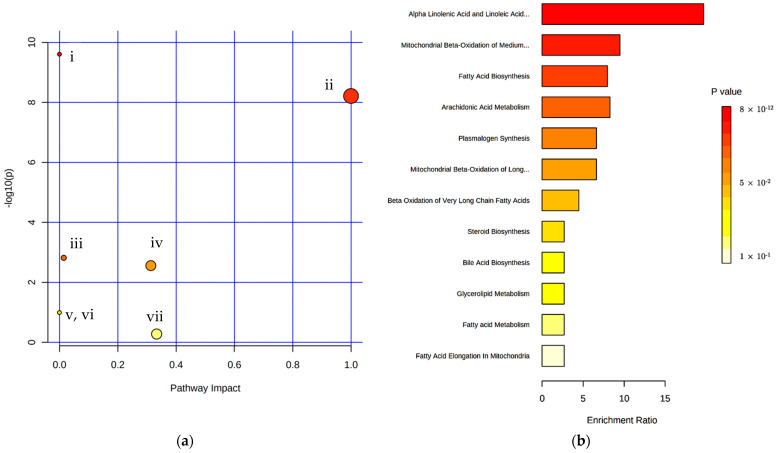
(**a**) Summary of pathway analysis: (i) Biosynthesis of unsaturated fatty acids; (ii) Linoleic acid metabolism; (iii) Fatty acid biosynthesis; (iv) Arachidonic acid metabolism; (v) Fatty acid elongation; (vi) Fatty acid degradation; (vii) alpha-Linolenic acid metabolism. (**b**) Quantitative enrichment analysis (QEA): Metabolite set enrichment overview. * Features 16, 38, and 39 (13-Octadecenoic acid, methyl ester; Cholesta-3,5-diene; Cholesta-2,4-diene) were excluded due to lack of metabolite ID (HMDB ID) conversion match.

**Table 1 metabolites-12-01111-t001:** Factors and levels tested for derivatization and extraction optimization using DoE. The optimized conditions are in bold.

Factors	−1	0	+1
T_1_ (°C)	**85**	95	105
t_1_ (min)	5	15	**25**
T_2_ (°C)	**85**	95	105
t_2_ (min)	5	15	**25**

**Table 2 metabolites-12-01111-t002:** Selected representative of each class for response optimization.

	SFA	MUFA	Omega-6	Omega-3	Omega-3
Analyte	C12:0	C14:1	C18:3 n-6	C20:4 n-3	C22:6 n-3
^1^t_R_ (min)	16.39	23.46	39.46	48.26	55.06
^2^t_R_ (s)	4.07	5.08	6.36	6.986	7.39

## Data Availability

The data presented in this study are available on request from the corresponding author. The raw data was obtained on LECO ChromaToF software (exported with baseline correction in cdf format) and analysed on GC Image software. In order to reproduce the results the user will require specific software. Therefore, to maintain the data integrity the data will be provided upon request.

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
