# Peer review of "Lipid Serum Profiling of Boar-Tainted and Untainted Pigs Using GC×GC–TOFMS: An Exploratory Study"

_metabolites, 2022, doi:10.3390/metabo12111111_

Round 1

Reviewer 1 Report

Journal: Metabolites (ISSN 2218-1989)

Manuscript ID: metabolites-2026315

Lipid serum profiling of boar-tainted and untainted pigs using GC×GC-TOFMS: An exploratory study

Kinjal Bhatt, Thibaut Dejong, Lena M. Dubois, Alice Markey, Nicolas Gengler, José Wavreille , Pierre-Hu- gues Stefanuto, and Jean- François Focant

Abstract: Mass spectrometry (MS) based techniques, including liquid-chromatography coupling, shotgun lipidomics, MS imaging, and ion mobility, are widely used to analyze lipids. However,
with enhanced separation capacity and an optimized chemical derivatization approach, compre-
hensive two-dimensional gas chromatography (GC×GC) can be a powerful tool to investigate some
groups of small lipids in the framework of lipidomics. This study describes the optimization of a
dedicated two-stage derivatization and extraction process to analyze different saturated and un-
saturated fatty acids in plasma by two-dimensional gas chromatography time-of-flight mass spec- trometry (GC×GC-TOFMS) using a full factorial design. The optimized condition has composite de- sirability of 0.9159. This optimized sample preparation and chromatographic condition was imple-
mented to differentiate between positive (BT) and negative (UT) boar-taint pigs based on fatty acid profiling in pig serum using GC×GC-TOFMS. A chemometric screening, including unsupervised
(PCA, HCA) and supervised analysis (PLS-DA), as well as univariate analysis (volcano plot) was performed. The results suggested that the concentration of PUFA ω-6 and cholesterol derivatives were significantly increased in BT pigs, whereas SFA and PUFA ω-3 concentrations were increased
in UT pigs. The metabolic pathway and quantitative enrichment analysis suggest the significant
involvement of linolenic acid metabolism.

It is a topic of interest to the researchers in the related area but the paper needs minor improvements before acceptance for publication. My detailed comments are as follows:

The introduction, materials and methods of the manuscript work very well, especially the part that corresponds to data processing. It would be interesting to include some additional data on the study sample. I'm talking about why that sample size, age, sex has been selected. (I couldn't find anything in the manuscript...). The number of samples is high enough to describe their nature.

The work seems to be of very good quality. Describes a complete workflow of analytical methods for analyzing lipids in plasma/serum The method also includes the liquid-liquid extraction of lipids from 25 μL of plasma/serum, optimized to maintain a broad selectivity towards multiple classes of FAMEs (SFA, MUFA and PUFA (ω-3), (ω-6)). I find extremely useful the description of an optimized sample preparation method, which is efficient for both human and porcine plasma. As the authors say, with the implementation of these applications, the possibility of translating this analytical protocol to any other study related to plasma/serum. As a final conclusion, I agree with the authors that the optimized method in this manuscript provides valuable information on the identification of FAMES without the need for detailed MS/MS investigations.

Author Response

Point 1: It would be interesting to include some additional data on the study sample. I'm talking about why that sample size, age, sex has been selected. (I couldn't find anything in the manuscript...). The number of samples is high enough to describe their nature.

Response 1:

Thanks for the positive feedback on our manuscript. Please find below the required information, and the part included in the revised manuscript.

As the main focus of this study was to establish an analytical workflow for lipid profiling of plasma/serum and to test the hypothesis if the method is efficient to distinguish boar tainted and untainted pig using it. Therefore, the sample size was kept at 40. In the upcoming part of the study, larger sample size will be investigated. The boar taint is mainly contributed by uncastrated male pigs [Reference 18]. Thus, selected all pigs were male. The age of pigs were 6 months ± 15 days.

The information were added to line 132*: (sex = male, age = 6 months ± 15 days).

* The mentioned line number in response is while keeping the all-markup option on while tracking the changes.

Reviewer 2 Report

Review of manuscript metabolites-2026315 entitled “Lipid serum profiling of boar-tainted and untainted pigs using GC×GC-TOFMS: An exploratory study”

Summary

In this manuscript, the authors present a GC×GC-TOFMS method to perform lipidomic profiling in combination with some data analysis tools. The optimized method is applied to characterize the lipid profile of boar-tainted pigs. Results highlight the relevance of PUFA w-6, w-3 and SFA as candidate markers.

Commentaries

In my opinion, the manuscript is mainly well-written, and the research presented in this manuscript is sound. Furthermore, most of the experimental data support the results and conclusions presented by the authors. Therefore, this manuscript could eventually be published in Metabolites. However, some issues should be corrected by the authors to improve the text's clarity and the results' reliability.

Line 54 – Monounsaturated?

Line 55 – The description of PUFA and the relevance of w-3 / w-6 should be improved.

Line 83 – Please, add a summary of the results in these previous works. In particular, give details of results obtained in references 21-23.

Line 111- How many samples of each type?

Line 143 – Include more details regarding the pre-processing methods applied to each used chemometrics tool.

Line 147 – Please, add Metaboanalyst reference.

Results – Move most of the description of the methods to the methods section.

Results – I feel that a Table summarizing the similarities/differences between the different data analysis tools could reinforce the presented results. Also, I think that it is necessary to provide a more in-depth description of the obtained results. For instance, C22:5 n-6 seems highly significant in the volcano plot but not in the VIPS values. Also, what happens with C14:0. Please, discuss.

Discussion/Conclusions – Summarize the benefits of the proposed approach when compared to the previous works in the literature. 
